# Data-driven understanding on soccer team tactics and ranking trends: Elo rating-based trends on European soccer leagues

**Dong Hee Jung**, **Jason J. Jung**\*

Department of Computer Engineering, Chung-Ang University, Seoul, Korea

\* j3ung@cau.ac.kr

**Data Availability Statement:** The dataset and source code used in this study are available at the following GitHub: https://github.com/kecau/SoccerTeamTrendAnalysis.

## Abstract

In modern soccer, strategy and tactics significantly impact team success. In this study, the application of these methodologies within the domain of association soccer is examined, with a particular focus on predicting team strategies via team trend analysis. Using a dataset comprising matches from the top five European soccer leagues, we analyzed team performance trends over time using the Elo rating system and rolling regression. Predicting strategies from soccer match datasets is a challenging. In our study, we propose methods based on count and rank to address these challenges. For the count-based method, the number of forwards, midfielders, and defenders is used to calculate respective defense and attack scores. For the rank-based method, teams are classified into various levels based on their rankings, and strategies are evaluated accordingly. This approach provides a more detailed perspective on strategic tendencies by considering team composition and performance at each level. Experimental results demonstrate the potential of our proposed methods to accurately identify and predict team strategies, offering significant implications for tactical decision-making in professional soccer. The findings indicate that the accuracy of predicting defensive strategies using count-based predictions was approximately 85%, while the performance of predicting aggressive strategies through rank-based predictions was 89%. Our methodology can be extended to the development of a predictive model aimed at forecasting team strategies, thereby assisting coaches with more effective preparation for upcoming matches.

## Introduction

The application of machine-learning and data analysis technologies in sports has increased significantly in recent years [1]. Various sports clubs are now leveraging big data for injury prevention and treatment as well as for optimizing training schedules and intensity for athletes [2, 3]. This trend extends beyond soccer, encompassing all sporting disciplines, highlighting the critical role of big data technologies [4]. Accordingly, in this study, emerging trends and strategic forecasting in association soccer were examined, with a particular focus on its implications and applications.

**Funding:** The author(s) received no specific funding for this work.

**Competing interests:** The authors have declared that no competing interests exist.

With its massive global popularity, association soccer continues to grow annually [5]. Major events, such as the FIFA World Cup, have significantly stimulated the soccer market through substantial capital investments [6]. The soccer industry has increasingly adopted big data along with various forms of data, such as images and videos, for market analysis and growth [7, 8]. Despite market expansion, soccer coaches' training and tactical decisions are often influenced more by personal judgment and experience than by data-driven insights [9]. The main goal of this research is to forecast team strategies based on trend analysis, thereby aiding coaches in their game preparation process, rather than directly predicting specific teams' tactics and game outcomes.

The advent of big data and artificial intelligence (AI) has profoundly transformed sports analytics [10, 11]. Recent advancements in big data analytics have paved the way for more detailed sports analyses, particularly in soccer [12–14]. There is an ongoing effort to surpass traditional methods by utilizing data science, data analysis, machine learning, and AI to examine team strategies and performances more thoroughly [15, 16]. This highlights the vital role of data science in sports, and demonstrates its potential to enhance competitive strategies through informed data-driven decisions [17]. The integration of these technologies provides unprecedented insights into the complexities of game dynamics, enabling the acquisition of actionable knowledge previously unattainable through conventional methods.

The Elo rating system, originally developed for ranking chess players, has been adapted in soccer to evaluate team performance dynamically. By updating team ratings after each match based on expected and actual outcomes, the system provides a nuanced understanding of a team's relative strength and performance trends over time. This approach offers a more comprehensive perspective compared to traditional ranking systems, making it highly relevant for analyzing team strategies and predicting future performance.

Constant technological advancements have introduced new methods into sports analytics, particularly in association soccer [18, 19]. Soccer, characterized by its dynamic and complex team interactions, requires more sophisticated analytical techniques as its commercialization grows. Existing studies often focused on team formation, position, and game records for analysis, overlook team trends [20]. The current study considered team trends as a fundamental characteristic used to predict team strategy. By concentrating on forecasting team trends and strategies, this study aimed to provide insights that can guide teams' proactive preparation and strategic responses in the highly competitive field of international soccer [21].

This study aimed to answer the following research questions:

- RQ1. What trends can be observed in a soccer team's performance over time?

- RQ2. Is it possible to group soccer teams into clusters based on their strategies?

- RQ3. Can a soccer team's strategy be predicted?

This study utilized data from the top five European soccer leagues, comprising 3,652 matches, to analyze using detailed metrics such as team formations, match outcomes, and rankings. The answers to these research questions contribute to the focus on trends and predictions of soccer team's strategies in sports analysis. To answer these questions, the following experimental design and research methodology are proposed:

- Develop methods to quantify and analyze trends within a soccer team. This approach involves a comprehensive analysis of previous games, providing flexibility for real soccer teams to apply in different scenarios.

- Apply clustering techniques to group teams with similar strategic approaches. This is conducted via two different methodologies for a more effective strategic similarity classification.

- Conduct experiments to predict the strategies of opposing teams in various situations. This predictive ability could assist team managers and coaches in their strategic decision making, potentially influencing the outcome of upcoming matches.

## Related work

As a foundation for sports analysis, it is crucial to explore related research in sports beyond soccer, notably basketball and baseball. These disciplines have experienced significant advancements in analytics, leveraging data to enhance team performance and strategic decision making. In basketball, analytics have led to the development of metrics such as the player efficiency rating (PER) and true shooting percentage (TS%), which provide insights into player contributions beyond traditional statistics [22]. Similarly, baseball has been transformed by sabermetrics, altering player valuations and strategic gameplay [23]. Recent studies have also explored the integration of machine learning in sports analytics, including its applications in soccer strategy optimization [24, 25]. Although these methods vary across sports disciplines, they share common objectives with soccer analytics. Cross-sport comparisons enrich the analytical field, underscoring the universal relevance of data-driven decision making in diverse sporting contexts.

This study focused primarily on team-based analysis rather than on individual player ratings or grades. In soccer analytics, the Elo rating system is essential for evaluating team trends and predicting match outcomes. Originally designed to assess the skills of chess players, the Elo rating system was adapted for soccer teams, providing a dynamic assessment of a team's strength over time [26]. By integrating the Elo rating system with machine-learning techniques and rolling regression methods, this study aimed to refine the analyses of soccer strategies [27].

In this study, clustering was employed by applying extracted key features. Clustering algorithms are utilized across domains such as image segmentation, object recognition, and data mining [28]. Specifically, a clustering algorithm for soccer formations assists in visualization, quantitative comparisons, and time-series analysis, aiding in the identification of team styles and positional exchange patterns [29]. Narizuka & Yamazaki (2019) [29] used a clustering algorithm to dynamically characterize soccer team formations across multiple matches, uncovering synergistic relationships via soccer player feature clustering. This approach leverages big data analysis techniques to examine soccer teams' tactics and formations [30]. The current study references such clustering-based studies and employs the K-means algorithm for clustering.

Traditional sports analytics has evolved beyond statistical methods to incorporate more sophisticated models, such as deep and reinforcement learning. Deep learning, through architectures such as convolutional neural networks (CNNs) and recurrent neural networks (RNNs), plays a crucial role in analyzing complex game footage and player movement patterns and offers insights into game dynamics [31]. In contrast, reinforcement-learning methods model decision-making in dynamic environments to optimize team strategies and player decisions through game scenario simulations [32]. Applying these technologies to soccer can facilitate analysis of team composition, player performance, and match outcome, mirroring the impacts observed in other disciplines.

Autoencoders, a type of neural network architecture designed for unsupervised learning, have demonstrated the potential for identifying meaningful features from complex datasets in sports analytics. Crucial for uncovering significant features within complex datasets without the need for explicit labeled outcomes, the introduction of the discriminative autoencoder by

Luo et al. (2019) [33] represents an important advancement in this field, merging the unsupervised learning capabilities of traditional autoencoders with supervised learning techniques [33]. This combined approach enables more effective feature extraction by utilizing available label information [34, 35]. Such advancements not only improve the feature representation quality but also show the adaptability of autoencoders to the nuanced and dynamic nature of sports data, paving the way for more accurate and predictive analyses in soccer.

## Soccer team trend analysis for league matches

In this section, we explore the application of the Elo rating system to analyze team trends and how rolling regression can be used to identify whether a team is on an upswing or downswing trajectory. The Elo rating system, which is fundamental in sports analytics, assigns numerical skill levels to teams based on their performance. After each match, the team score is updated to reflect the latest results. Applying Elo ratings offers deeper insights than traditional rank-based trend analyses, capturing nuanced changes in team performance that rankings alone may not reveal.

To demonstrate this, we present case studies of several teams that have implemented Elo ratings and rolling regression. These examples elucidate how team trends are identified, showcasing both the upswings and downswings. Understanding these dynamics is crucial for understanding the fluid nature of team performance in league play and provides strategic value to coaches, analysts, and enthusiasts alike.

## Team rank for Elo rating

The Elo rating system, originally designed for chess player evaluation, has found widespread application in various competitive sports. Notably, it has been adapted to assess team performance in soccer, where it serves as a crucial algorithm for adjusting team ratings based on match results and statistically estimating relative skill levels. This section outlines the fundamental aspects of the Elo rating system and its modifications for soccer applications.

In this study, we applied the Elo rating system to soccer teams rather than individual players. This adaptation necessitates an approach that focuses on evaluating the collective team performance. The core principle of the Elo rating system, as detailed in [36], involves analyzing match results to statistically measure a team's current strength, considering both past performances and recent outcomes.

The Elo rating calculation in soccer hinges on the principles of expected and actual match outcomes. The expected team score is given by:

$$E_A = \frac{1}{1 + 10^{(R_B - R_A)/D}} \tag{1}$$

where $E_A$ represents Team A's expected score in a match with Team B. $R_A$ and $R_B$ are the current Elo ratings of Teams A and B, respectively. The divisor $D$, typically set at 1500 in soccer, normalizes the difference in rating, ensuring a balanced comparison between teams.

After a certain match, the Elo ratings are updated to reflect the actual results, as follows:

$$R'_A = R_A + K \times (S_A - E_A) \tag{2}$$

where $R'_A$ indicates Team A's post-match updated rating. The K-factor $K$ determines the extent of rating adjustment based on the match outcome. Iteratively applying these equations after each match ensures that team ratings are continuously updated, reflecting both consistent

performance and occasional deviations.

$$S_A = \begin{cases} 1, & \text{Team win} \\ 0.5, & \text{draw} \\ 0, & \text{otherwise} \end{cases} \qquad (3)$$

The actual result achieved by Team A is denoted by $S_A$, where $S_A$ represents the actual match result. Refusing to give one point for a game you need to win, or 0.5 points for a draw, will burden with zero points [37]. This methodology offers a more comprehensive view of a team's capabilities and adaptability over time. In this study, the Elo rating system was utilized not only as a ranking tool but also as an insightful metric for analyzing a team's strategic and performance trends.

Fig 1 shows the application of the Elo rating system to Serie A teams. Although we calculated Elo ratings for 98 teams, for clarity, we present the ratings for four teams representing high, mid-high, mid-low, and low levels. This selection provides insight into the effectiveness of the system and the varied standing of teams in a competitive league, the Italian top-flight in this example. The current FIFA Women's World Rankings utilizes a modified version of the Elo formula. Our study revealed that the system's utility extends beyond traditional rankings, enabling the exploration of team strategies and their evolution.

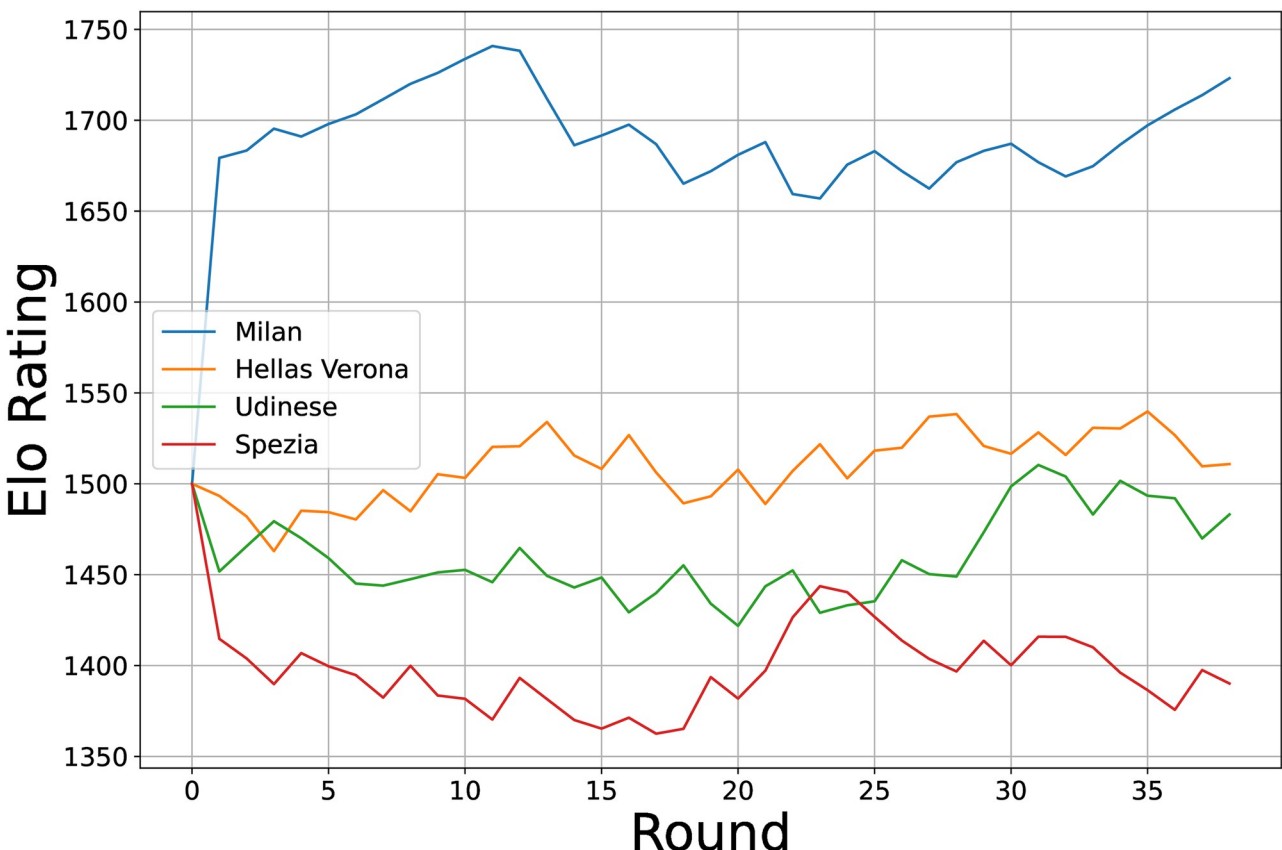

**Fig 1. Elo ratings of Serie A teams.**

In conclusion, the mathematical sophistication and adaptability of the Elo rating system make it an indispensable analytical tool in soccer. Its application broadens the ability to discern subtle shifts in team performance, offering a detailed perspective on the dynamic competitive soccer landscape.

## Rolling regression for Elo rating trend

In sports analytics, a critical aspect is understanding and analyzing team performance over time. This section presents the application of rolling regression to the Elo rating system and offers novel insights into soccer teams' performance trends. Unlike moving averages, which smooth the data to provide a general direction, rolling regression discerns the direction and magnitude of trends through the slope of a fitted linear model [38].

In this study, we used the Elo rating to calculate a soccer team's score based on match results and updated the Elo score after each match, considering the expected and actual results. Subsequently, rolling regression was applied to these ratings to analyze the performance trends. The choice of window size in rolling regression is crucial; a smaller window may capture noise, whereas a larger window may obscure short-term fluctuations. In our analysis, a window size of 5 was selected for optimal balance. Through the iterative application of rolling regression to the Elo ratings, we conducted a regression analysis for each team across matches and calculated the regression line slope for each window. Then, this slope was used to categorize the performance trends as 'Upswing' or 'Downswing.'

Fig 2 illustrates these trends. For example, the Elo ratings of Milan, Hellas Verona, Udinese, and Spezia are plotted against matchdays, with trend lines from the rolling regression superimposed. This visualization provides a clear insight into each team's performance trajectory throughout the season. Although both rolling regression and moving averages analyze trends, they differ significantly in their approach and interpretation. While smoothing data, moving averages can obscure nuanced changes. In contrast, rolling regression, by providing a slope or rate of change, offers a more precise and quantifiable trend measure, enhancing the depth of analysis of team performance trends.

In conclusion, the integration of rolling regression with the Elo rating system has advanced our understanding of soccer team performance. This approach not only highlights current standings but also elucidates significant results and patterns, thereby enriching the Elo rating system's utility from static rankings to dynamic trend analysis, offering a deeper understanding of team performance in European soccer leagues.

## Identifying upswings and downswings

Addressing RQ1 (What trends can be observed in a soccer team's performance over time?), in this subsection, we focus on the application of rolling regression to the Elo rating system to identify performance trends in soccer teams. Analyzing these trends is crucial for understanding the ups and downs in team performance, offering a more dynamic perspective than traditional ranking systems.

After applying the rolling regression to the Elo rating system, we focused on identifying and interpreting the ups and downs in team performance. Such trends are pivotal in sports analytics, as they offer deeper insights into a team's dynamics throughout the season. Unlike traditional rankings, the Elo-based approach addresses limitations by considering the strength of opponents. The methodology involves analyzing the slopes derived from the rolling regression of Elo ratings. A positive slope indicates an upswing or successful performance period, while a negative slope indicates a downturn in team performance. The trends of 98 teams were extracted using this approach.

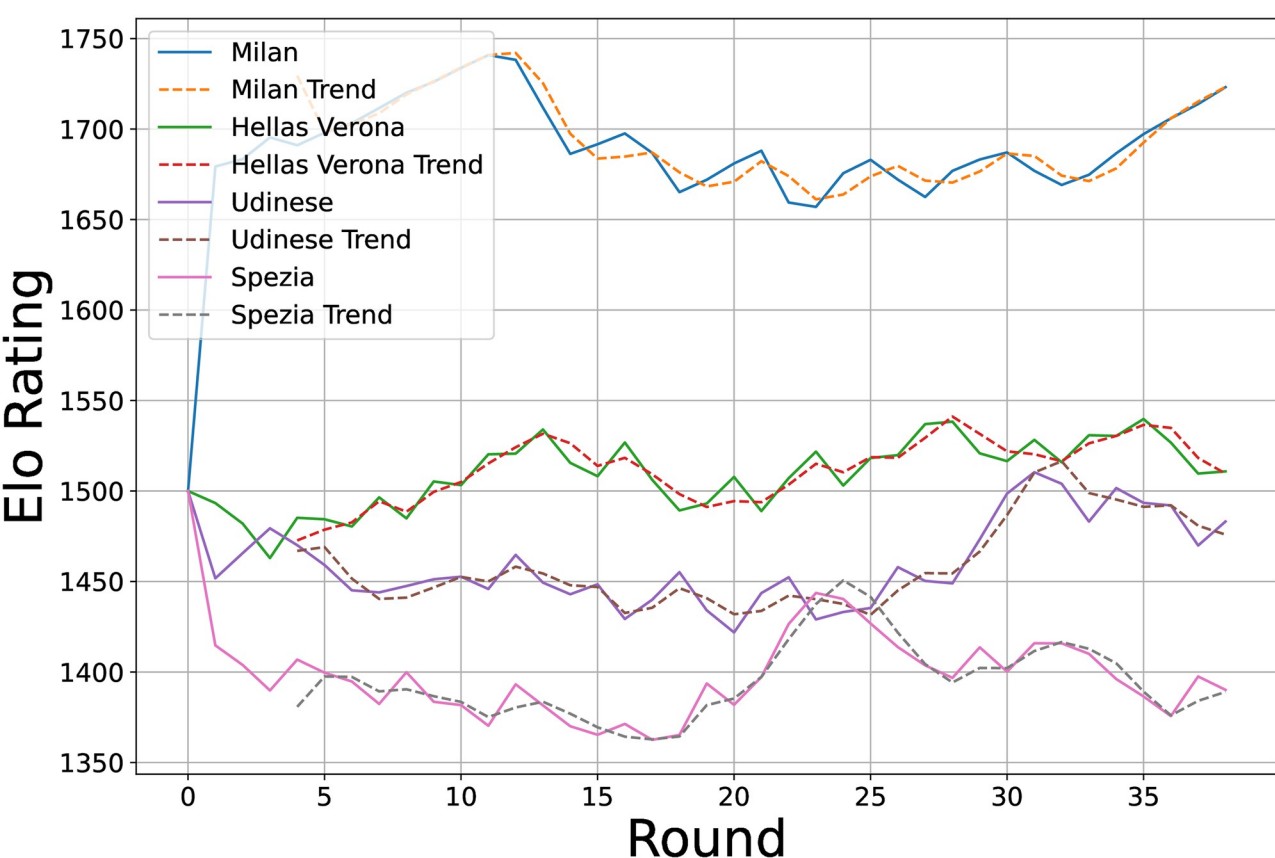

**Fig 2. Rolling regression applied to the Elo rating of four Serie A teams.**

Fig 3 illustrates these trends by using a rolling regression based on the Elo rating system. The figure depicts the Elo ratings and rolling regression trends for the four representative Serie A teams, highlighting the upswings (green dots) and downswings (red dots). This visualization elucidates team trends over a season, demonstrating that changes in Elo scores alone do not necessarily indicate definitive upswings or downswings.

Understanding these trends is invaluable for the team management and coaching staff. This analysis helps identify effective strategies and periods that necessitate tactical changes. It also informs decisions regarding training, player substitutions, and strategy development. This analytical approach, which links statistical analyses with sports strategies, offers new insights into team performance over time. Identifying performance trends through rolling regression provides sports analysts and strategists with a dynamic understanding of team performance in competitive soccer leagues beyond traditional experience-based management.

These experiments clarify that the optimal number of games for trend analysis varies significantly among teams. This finding highlights the importance of a customized approach for soccer strategy analysis. Additionally, it was found that the number of games to be considered in the analysis can be determined based on team level and recent performance.

These results suggest that as the season progresses, recent matches have a greater impact than previous matches. The proposed method allows trends to influence the analysis more significantly by reflecting the results of each match in terms of team strategy and performance. This approach, which gives greater weight to recent matches, is particularly useful for

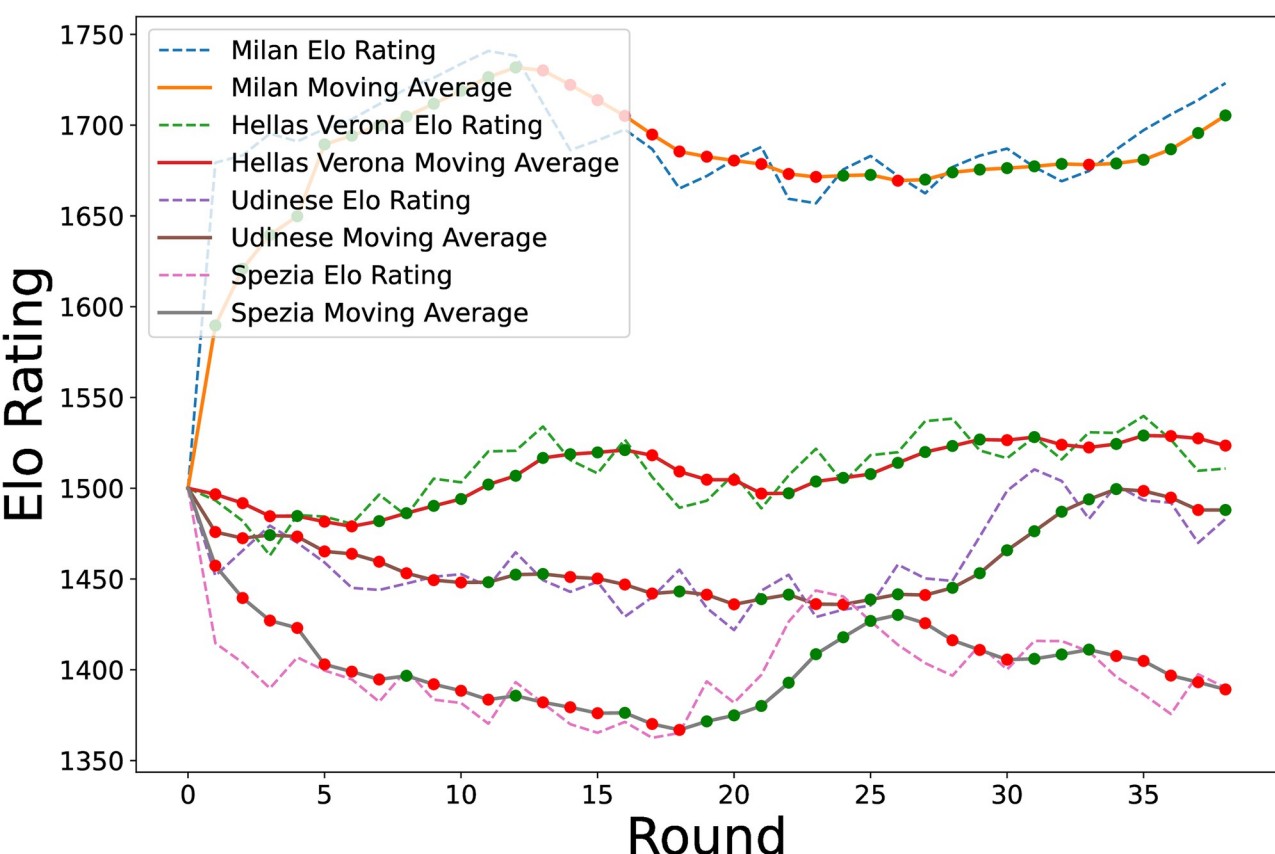

**Fig 3. Elo ratings and rolling regression trends for four Serie A teams.**

predicting future game strategies and can be an essential tool for coaches and analysts in their decision-making process.

## Predictive modeling of team formation strategies

In this section, the methodologies modeled to determine a team's formation strategy are outlined. We utilized two primary approaches to investigate team strategy: the count- and rank-based approaches. These methods provide various perspectives and critical insights into team formations and their impact on game strategies, aiding in the analysis and prediction of team strategies.

To determine the team strategy, we performed feature extraction using an autoencoder. Because there were no correct answer labels, labels for the offensive and defensive formations were assigned through clustering. Finally, we created a prediction model for the team's strategy using clustered labels using a multi-layer perceptron (MLP).

## Team strategic approach for formation style

In soccer, match tactics refer to the strategic formations and play styles employed by a team to achieve specific objectives during a game. Among these tactics, the composition of the defensive line significantly influences a team's strategic direction and game management.

The 3-back formation consists of three central defenders and two wing-backs, designed to strengthen central defense while allowing wing-backs to support both offensive and defensive

plays. This formation effectively utilizes wide spaces on the flanks and facilitates quick counter-attacks. Common formations within this system include 3-0-5-0-2 and 3-0-4-0-3.

The 4-back formation is the most stable system, comprising four defenders to maintain a balance between defense and attack. Full-backs in this system contribute to both defensive coverage and offensive support, offering high tactical flexibility that adapts to various game scenarios. Representative formations include 4-0-4-0-2 and 4-0-3-0-3.

The 5-back formation prioritizes defensive solidity by deploying five defenders to fortify both central and wide defensive zones. While highly effective in maintaining a robust defense, this system may limit offensive build-up. Typical formations include 5-0-3-0-2 and 5-0-4-0-1.

To explore team strategies, we employed two primary approaches: the count-based approach, which simplifies team dynamics analysis by focusing on player positions, and the rank-based approach, which emphasizes the outcomes of matches in the strategic analysis.

The count-based approach emphasizes a straightforward analysis of team formation, typically involving defenders, midfielders, and attackers. For example, in a 4-0-4-1-1 formation, the attack score based on one forward and five midfielders is six, while the defense score considering four defenders and five midfielders is nine. This method assesses a team's offensive and defensive strengths based on player positions and provides a clear metric for evaluating a team's inclination toward offensive or defensive styles.

Building on the foundation of the count-based approach, the rank-based approach categorizes teams into various levels based on their ranking and evaluates their strategies accordingly. This approach offers a more detailed perspective on strategic inclinations by considering team formation and performance at their respective levels. To effectively implement this approach, teams are categorized into 'High,' 'Mid-High,' 'Mid-Low,' and 'Low' levels based on their league ranking. This categorization, as detailed in Table 1, is critical for assessing the teams strategic positioning and performance.

Within each level, we calculated the average goals for (GF) and against (GA) for each team formation. The teams were ranked within their respective levels based on these averages. Additionally, percentile scores were computed to provide a comparative metric across levels. This methodology allows for a comprehensive team analysis, emphasizing their relative strengths and weaknesses in offensive and defensive aspects, depending on their ranking and formation style. Table 2 lists the offensive and defensive rankings of the different formations at each level.

Integrating the count- and rank-based approaches not only provides immediate analytical benefits but also lays the groundwork for developing more dynamic and responsive team strategies. This comprehensive view aids coaches and analysts in their strategic planning and decision making [9].

## Match-result-based strategy feature extraction

In this section, the methodology for extracting features from match results is investigated, which is crucial for enhancing our understanding of team strategies. Initially, we revisited match data processing using the previously discussed count- and rank-based approaches.

**Table 1. Team level.**

| Team Level | Ranking |
|---|---|
| High | 1–5 |
| Mid-High | 6–10 |
| Mid-Low | 11–15 |
| Low | 15–20 |

**Table 2. Ranking-based formation rank.**

| Team Formation | Level | Offensive Rank | Defensive Rank |
|---|---|---|---|
| 3-0-4-0-3 | High | 5 | 8 |
| 3-0-4-0-3 | Low | 10 | 5 |
| 3-0-4-0-3 | Mid-High | 13 | 8 |
| 3-0-4-0-3 | Mid-Low | 7 | 10 |
| 3-0-4-1-2 | High | 13 | 11 |
| 3-0-4-1-2 | Low | 3 | 11 |
| 3-0-4-1-2 | Mid-High | 6 | 9 |
| 3-0-4-1-2 | Mid-Low | 6 | 17 |

These methods enable us to understand team strategies and provide fundamental insights by examining simple scoring extractions from team formations. For instance, in the 4-0-4-1-1 formation, we calculated an attack score of 6 and a defense score of 9. The feature extraction process, which leverages both numerical and categorical features, enhanced the depth and precision of the analysis.

This process is further refined by integrating the rank-based approach, in which teams are classified into various levels based on league rankings. This classification allows for a more detailed analysis of offensive and defensive capabilities, surpassing the simplicity of the count-based scoring mechanism.

Autoencoders play a crucial role in transforming high-dimensional data into a more manageable form, thereby facilitating an in-depth analysis of team strategies. The transformed data are then clustered using the K-means algorithm, effectively categorizing team formations into offensive and defensive styles. This method addresses the lack of definitive labels for team formation strategies and is critical for a comprehensive understanding of team strategies [39].

The encoding and decoding processes in our neural network, which includes one hidden layer, are represented as follows:

$$Y = f_\theta(X) = s(WX + b_X) \tag{4}$$

$$X' = g(Y) = s_g(W'Y + b_Y) \tag{5}$$

In Eq (4), $Y$ represents the encoded feature set obtained from input data $X$. Function $f_\theta$ includes the weights $W$ and biases $b_X$ of the encoder, where $s$ is the activation function. This process compresses original high-dimensional data into a lower-dimensional representation. Eq (5) expresses the decoding function, where $X'$ represents the reconstructed data. Function $g$ uses the weights $W'$ and biases $b_Y$ of the decoder, where $s_g$ is the activation function. This phase attempts to reconstruct the original data from the encoded representation, which is essential for loss minimization during training.

K-means clustering played a vital role in generating clear labels for the offensive and defensive formations. This approach is essential to thoroughly understand team strategies. By applying this methodology, distinct team formation and strategic patterns emerged. Specific formations were more likely to be associated with offensive or defensive playing styles. The use of these mathematical formulations in our neural network model enables a nuanced understanding and categorization of team strategies. These insights significantly contribute to the accuracy of subsequent predictive models in soccer strategy analyses.

## Results

### Experimental settings

Our goal was to gather and analyze key information about team tactics from the top five European soccer leagues, namely, England, Spain, France, Italy, and Germany. We collected data from various websites, including sports-reference(https://www.sports-reference.com/), covering 3652 games played by 98 teams during a single season. The data that we focused on included aspects such as matchday, team name, opposing team, team formation, opponent formation, match result, home/away status, GF, GA, and team rank. Preprocessing involved normalizing numerical variables such as goals for and against, encoding categorical features such as formations and home/away status, and addressing missing data by interpolating results based on recent team performance. We focused on league matches because they provide consistent data, and decided not to include matches from continental club tournaments or domestic cup competitions.

We chose major European soccer leagues that are well-known and popular worldwide. This selection was important for our aim, which was to understand team strategies and trends by analyzing their performance in head-to-head league matches. Each team's participation in both home and away matches was crucial to the analysis. Table 3 lists the leagues and countries from which we gathered our data.

Furthermore, to add depth to our analysis, we also used worldfootball(https://www.worldfootball.net/) to obtain real-time team rankings after each matchday. This extra information was vital for identifying trends, such as upswings or downswings in team performance based on their changing ranking, thereby offering a more comprehensive view of how teams perform over time.

All analyses and model computations were performed using Python. The parameters set during training for each proximity are as follows: batch size = 128 and epochs = 10. We used the Adam optimizer with a dropout rate of P drop = 0.1 and a learning rate of 0.01. Dropout was applied to the output of each sub-layer, and L2 regularization (weight decay) of 0.001 was applied in the fully connected layers.

### Formation style clustering result

In this subsection, by addressing RQ2 (Is it possible to group soccer teams into clusters based on their strategies?), we explored the clustering of soccer teams. This analysis aimed to identify groups of teams with similar strategic trends, both in terms of our focus team and their opponents, as well as in their offensive and defensive capabilities. A particular case study was conducted involving matches in which a lesser-ranked team emerged victorious over a higher-ranked opponent. The case study concentrated on such matches to discern patterns in the strategies of underdogs overcoming favored teams.

The dataset, comprising 108 instances in which a weaker team defeated a stronger one, was segmented into two halves for detailed analysis. The first half spanned from Matchday 1 to the

**Table 3. European leagues and countries used to collect data.**

| League | Country |
|---|---|
| La Liga | Spain |
| Premier League | England |
| Bundesliga | Germany |
| Serie A | Italy |
| Ligue 1 | France |

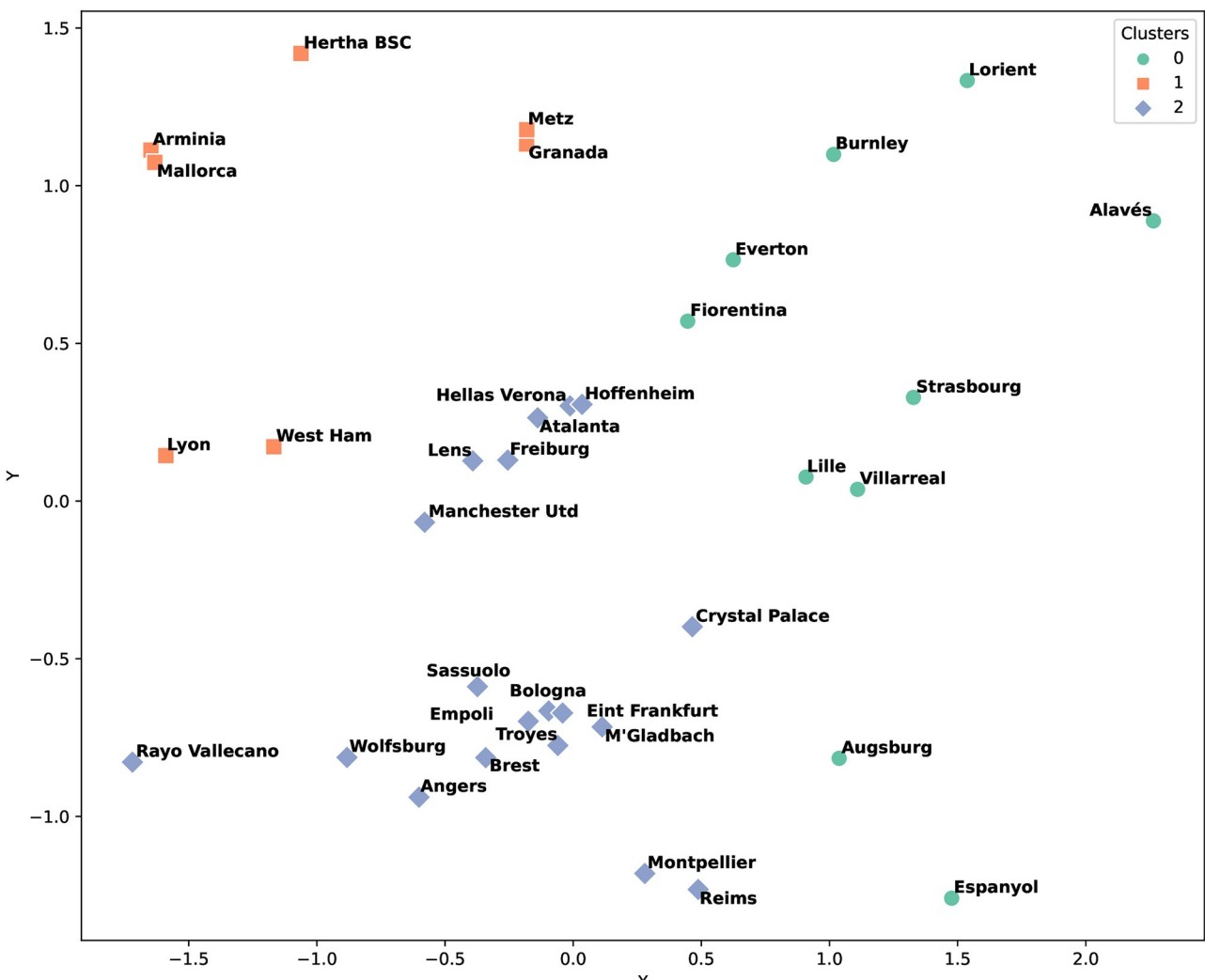

**Fig 4. Clustering count results for the first half of the European soccer season.**

midpoint of Matchday 19. Fig 4 shows the clustering results for this period, revealing discernible clusters of teams with similar strategic approaches. The analysis, based on three distinct clusters, allowed for an insightful delineation of the strategic styles prevalent among teams.

We examined the tactics primarily used in each cluster. Cluster 0 frequently employed formations such as '4-0-4-0-2' and '5-0-3-0-2'. Cluster 1 predominantly used the '4-0-2-3-1' formation. Teams in Cluster 2 mainly utilized formations like '4-0-3-0-3' and '3-0-4-0-3'. Each cluster also exhibited significant characteristics beyond tactics. For instance, Cluster 0 mainly relied on defensive strategies, evidenced by the highest defense score of 8 points according to our calculation method. Conversely, Cluster 2, with a defense score of 7 points, played more offensively compared to Cluster 0. The difference between Cluster 1 and Cluster 2 arises more from the tactical variations rather than the overall play style.

Fig 5 shows the clustering results for the second half of the season, Matchdays 20 to 38. Clusters 1 and 2 comprise teams with nearly identical strategic profiles. Notably, "M'Gladbach" maintained cluster grouping across both halves, indicating a consistent strategic

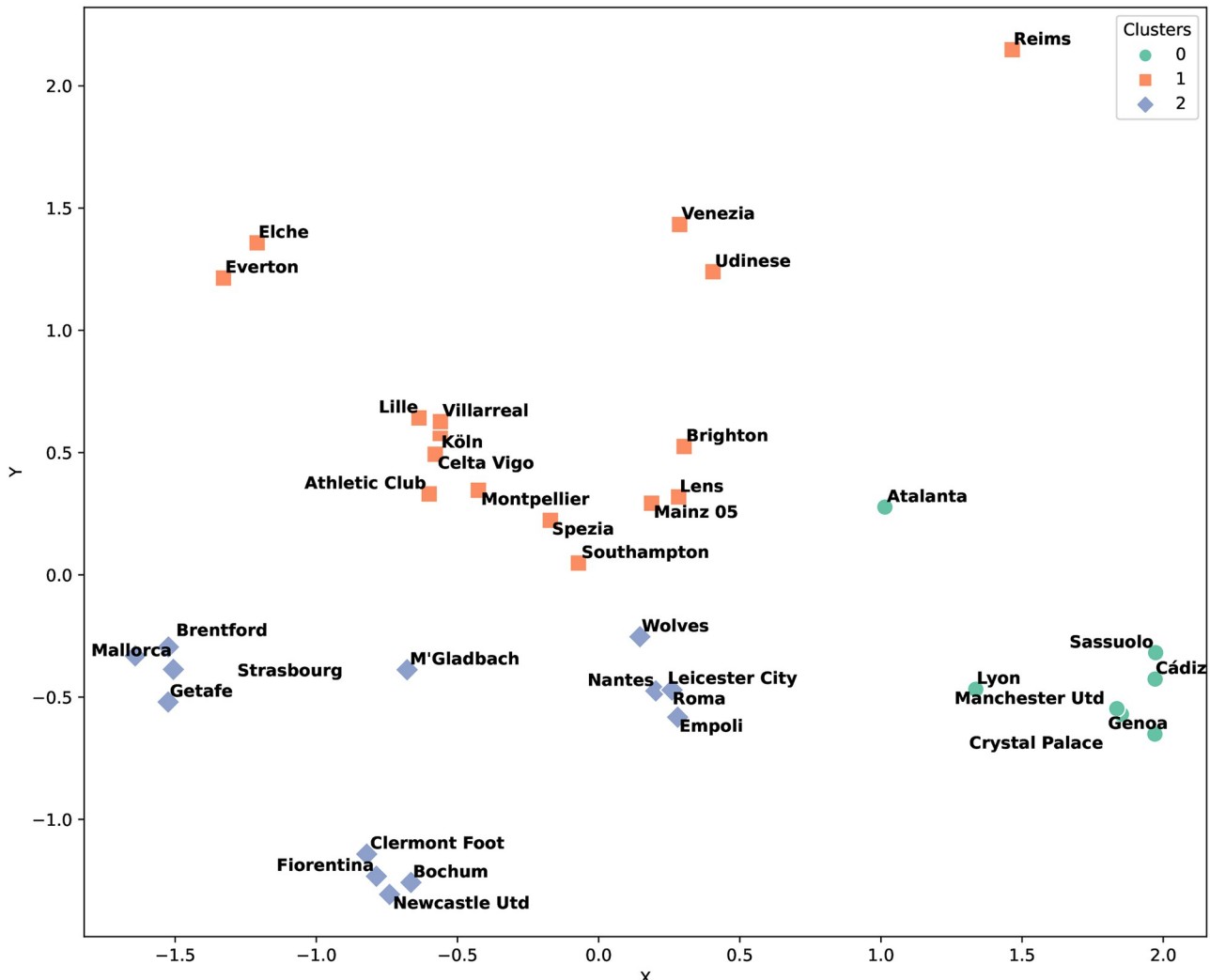

**Fig 5. Clustering count results for the second half of the European season.**

approach. Conversely, "Crystal Palace" underwent a strategic transition, moving from Cluster 2 in the first half to Cluster 0 in the second half, indicating a shift in strategy to better challenge stronger teams.

The ranking method offers a different perspective, as shown in Figs 6 and 7. This approach resulted in a more even distribution of teams across clusters, suggesting a general strategic similarity. The ranking method revealed that Clusters 0, 1, and 2 contained teams that utilized a broad spectrum of strategies.

The clustering performed in this study faces several challenges due to its reliance on a single-season dataset, which limits its analysis to a small subset of matches. The focus on matches where weaker teams beat stronger ones further narrows the study's scope to a selected number of teams. Future research should aim to overcome these limitations by expanding the dataset to enable a more comprehensive analysis that includes a broader variety of teams and strategic dynamics.

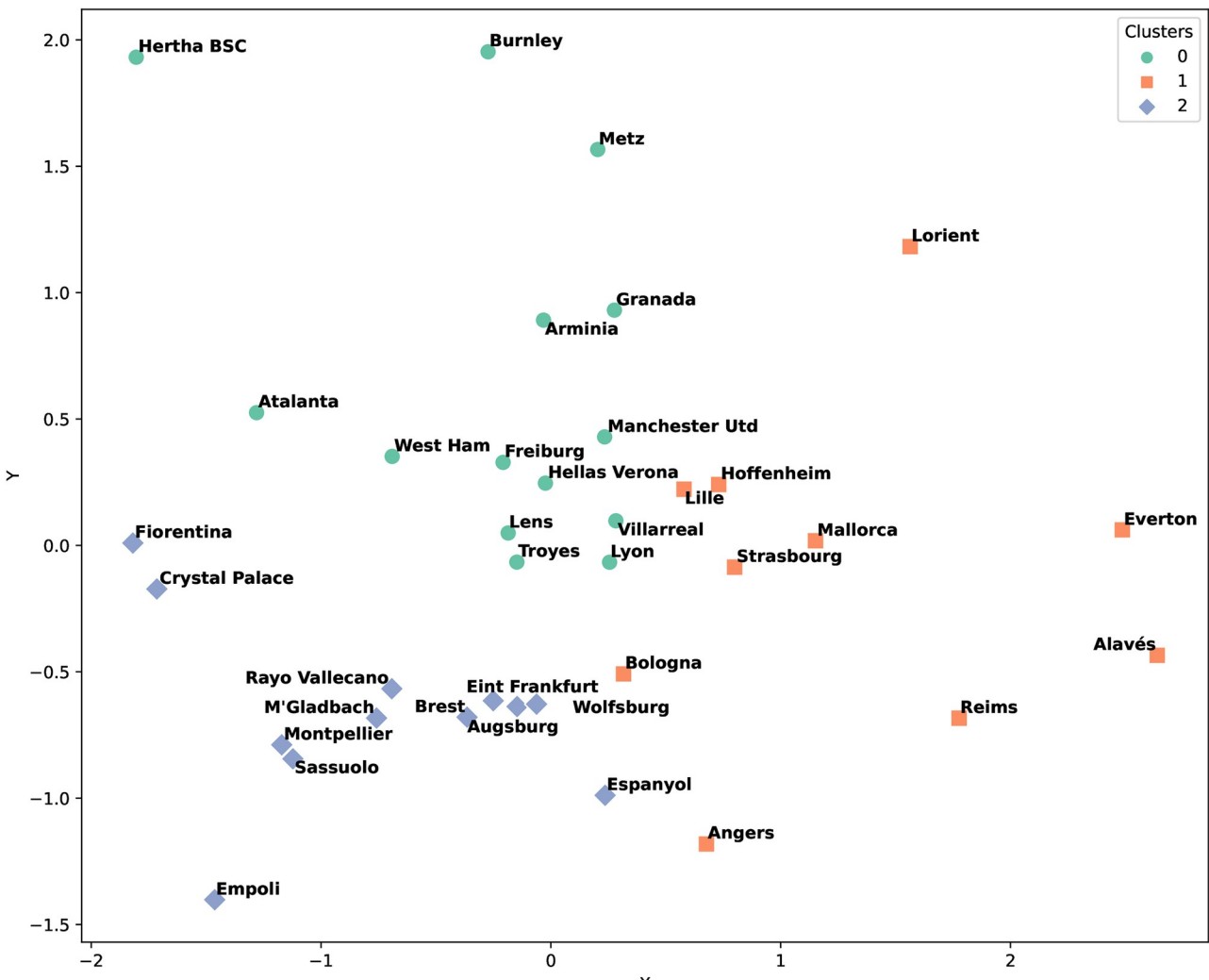

**Fig 6. Clustering ranking results for the first half season of the European soccer.**

### Formation style prediction

In this subsection, addressing RQ3 (Can a soccer team's strategy be predicted?), the team strategy forecasting results are presented by comparing various classification models. To predict the strategies employed by soccer teams, we trained several classification models on our dataset and evaluated their performance in terms of accuracy, precision, and recall on a test dataset. We utilized K-fold cross-validation to assess the models' performance. This method involves dividing the data multiple times for training and evaluation, thereby ensuring that our performance metrics reflect the models' generalized capabilities, rather than their performance in a single data partition.

In this study, the precision and recall metrics for predicting soccer team strategies served to measure how accurately a model could predict a specific strategy. For instance, if a model predicts that a team will employ an aggressive strategy, the precision metric assesses the accuracy of this prediction in matching teams that employ aggressive strategies. Recall measures the model's ability to correctly identify teams employing a particular strategy from among all

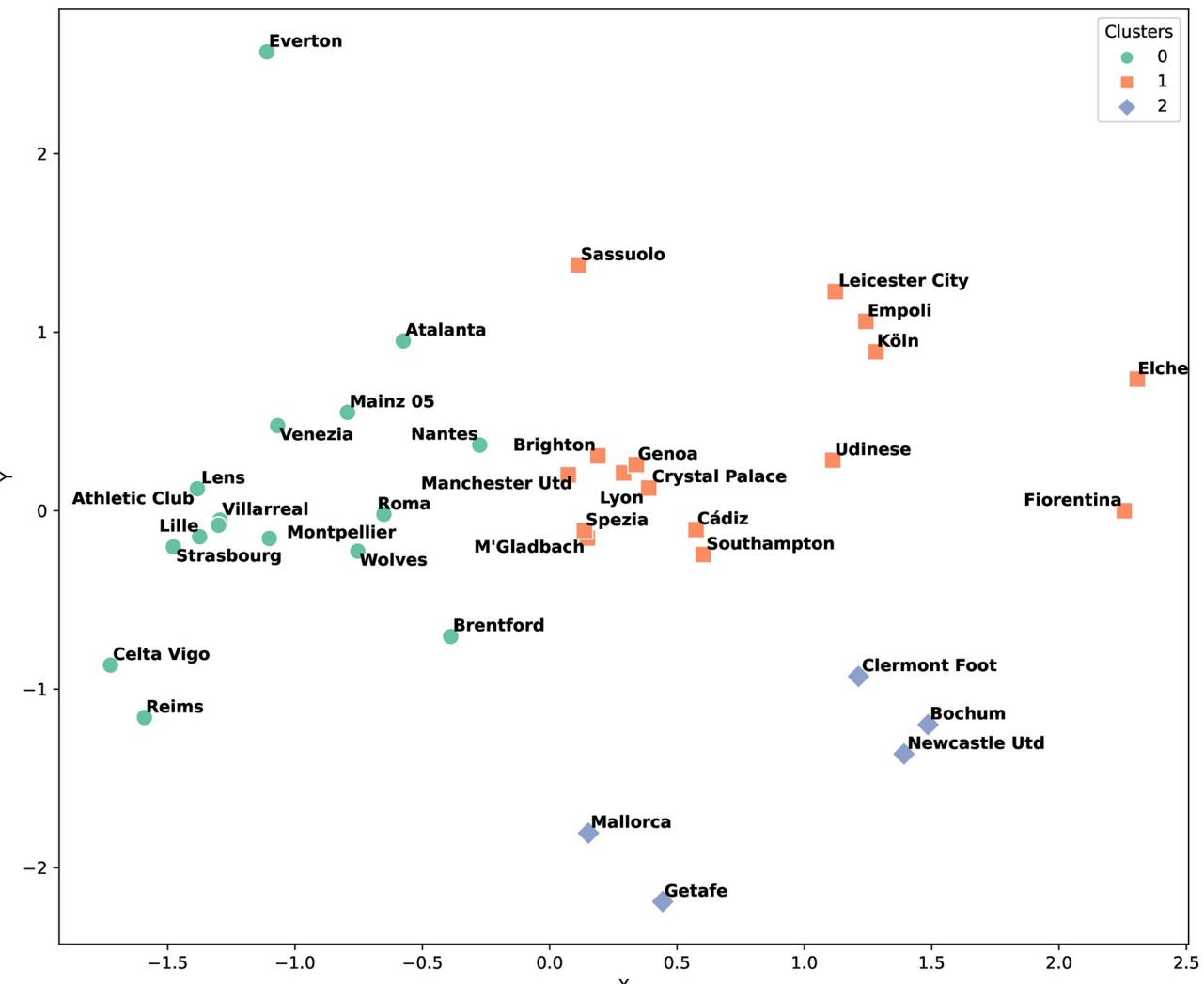

**Fig 7. Clustering ranking results for the second half season of the European soccer.**

teams that the model predicts as doing so. Thus, in the context of soccer team strategy prediction, precision refers to the proportion of teams correctly identified as using a particular strategy out of all teams predicted as such, while recall indicates the proportion of teams correctly predicted as using a strategy out of all teams that employ that strategy. The analysis of a soccer team's strategy requires careful assessment of the tradeoff between failing to identify a particular strategy and incorrectly identifying one, guiding the choice of the model based on whether precision or recall is deemed more critical.

To analyze the impact of team formation styles on match outcomes, we employed several predictive models: MLP, Random Forest, Gradient Boosting, Logistic Regression, and Decision Tree. Each model was selected for its unique characteristics and strengths. MLP is effective at capturing nonlinear relationships in complex datasets. Random Forest enhances robustness by combining multiple decision trees and managing datasets effectively. Gradient Boosting builds trees sequentially, correcting errors from previous iterations, thereby improving accuracy in complex scenarios. Logistic Regression offers a straightforward approach for

**Table 4. Count-based formation prediction.**

| Model | Accuracy | Precision | Recall |
|---|---|---|---|
| MLP | 0.62 | 0.62 | 0.68 |
| Random Forest | 0.62 | 0.63 | 0.63 |
| Gradient Boosting | 0.64 | 0.64 | 0.71 |
| Logistic Regression | 0.53 | 0.55 | 0.52 |
| Decision Tree | 0.62 | 0.63 | 0.61 |

binary classification, providing insights into probabilistic outcomes and feature influences. The Decision Tree model is intuitive and interpretable, making it useful for understanding decision processes and modeling feature interactions. For comprehensive performance evaluation, these models were assessed using cross-validation and metrics such as accuracy, precision, recall, and F1 score.

Table 4 lists the evaluation metric values of the count-based strategy predictions. For count-based formations, the number of players in specific positions significantly affects the prediction outcome. Gradient boosting achieved the highest accuracy, precision, and recall values in the count-based strategy prediction, demonstrating a superior predictive performance. Conversely, logistic regression exhibited the lowest performance.

Table 5 lists the evaluation metric values of the ranking-based strategy prediction. In ranking-based formations, factors such as team level, GF, and GA were found to be the most influential. Similarly, gradient boosting exhibited the highest accuracy in the ranking-based strategy prediction, with excellent precision results, while random forest exhibited the best recall performance. Overall, ranking-based predictions yielded higher average accuracy values and slightly better performance than count-based predictions.

The ranking-based approach incorporates contextually relevant features, such as team level, GF, and GA, offering more comprehensive insights into a team's capabilities and strategic style, thereby enabling models to leverage a richer dataset. The consistent success of gradient boosting across both prediction methods underscores its efficacy in modeling complex patterns and interactions between features. In the ranking-based approach, the random forest model achieved a recall rate of 0.72, the highest among the models tested. However, this also indicates a 28% inaccuracy. While this demonstrates the random forest model's relative strength in classification compared to other models, it also highlights the need for further improvements to enhance accuracy. This suggests that there is still room for improvement in this model.

The observed differences in model performance between the count- and ranking-based methods can be attributed to each model's specific approach to processing and learning from strategic features. Gradient boosting's across-the-board success highlights its versatility in adapting to simple positional data and complex contextual information. The disparity in

**Table 5. Ranking-based formation prediction.**

| Model | Accuracy | Precision | Recall |
|---|---|---|---|
| MLP | 0.68 | 0.70 | 0.64 |
| Random Forest | 0.74 | 0.75 | 0.72 |
| Gradient Boosting | 0.78 | 0.82 | 0.71 |
| Logistic Regression | 0.56 | 0.55 | 0.55 |
| Decision Tree | 0.75 | 0.77 | 0.70 |

performance between the two prediction methods emphasizes the importance of data representation for accurately modeling soccer strategies. Count-based data offer a straightforward, albeit simplified, perspective on team composition, whereas ranking-based data provide a more nuanced and contextually rich understanding, thereby enabling broader strategic insights.

The variation in model efficiency suggests a nuanced alignment between the model capabilities and the predictive tasks. Models capable of intricately managing feature complexity and interactions, such as gradient boosting, excelled in both scenarios. In contrast, models such as logistic regression may struggle with the more complex and less linear nature of ranking-based data. Based on these findings, we discuss not only the predictive capabilities of various models but also the intrinsic nature of soccer strategies. This analysis suggests that, while count-based metrics provide valuable insights, integrating ranking-based metrics offers a more comprehensive view of a team's strategic approach.

## Model evaluation

In this section, we evaluate the performance of predictive models designed to forecast the characteristics of sports team performances based on their formations and past match statistics. The dataset was subjected to comprehensive preprocessing to optimize the modeling. This included the normalization of numerical features, such as GF, GA, team rankings, and calculated attack and defense scores, as well as the encoding of categorical variables representing team formations and match outcomes. Feature engineering was pivotal in establishing the foundation for our predictive models and enhancing their learning capability using the dataset.

Our MLP model integrates both numerical and categorical inputs. Numerical features were subjected to linear transformation, whereas categorical features were processed through embedding layers to capture the intrinsic relationships within the data. The architecture comprised dense layers with ReLU activation and a final sigmoid layer for binary classification, assuming that the target variable was binary. Concurrently, an Autoencoder model was deployed to distill high-dimensional data into a more manageable lower-dimensional space, thereby facilitating a more efficient clustering process. The encoder segment aims to compress the input data, while the decoder seeks to reconstruct the input using a compressed representation. By applying these methodologies, we comprehensively evaluated the team strategies.

The dataset was divided into training and testing subsets, with 80% of the data allocated for training. Both models were compiled using the Adam optimizer, with the MLP model focusing on binary cross-entropy loss for classification and the Autoencoder model focusing on the mean squared error for reconstruction fidelity. The MLP model showed promising accuracy levels, effectively predicting whether a team formation was offensive or defensive. The confusion matrix provided deeper insights into the model performance, highlighting its precision in classifying true positives and negatives. The normalized values facilitate an easy understanding of the model's predictive consistency across classes, playing a pivotal role in assessing the model's practical utility in sports analytics.

Fig 8 presents the confusion matrix based on count-based data, which reveals better accuracy in predicting defensive strategies than offensive strategies. Conversely, Fig 9 presents the confusion matrix based on ranking-based data, indicating superior accuracy in forecasting offensive strategies compared with defensive strategies. Both methodologies exhibit distinctive advantages: one excels at predicting offensive strategies, whereas the other is proficient in forecasting defensive strategies, each with its inherent strengths and weaknesses.

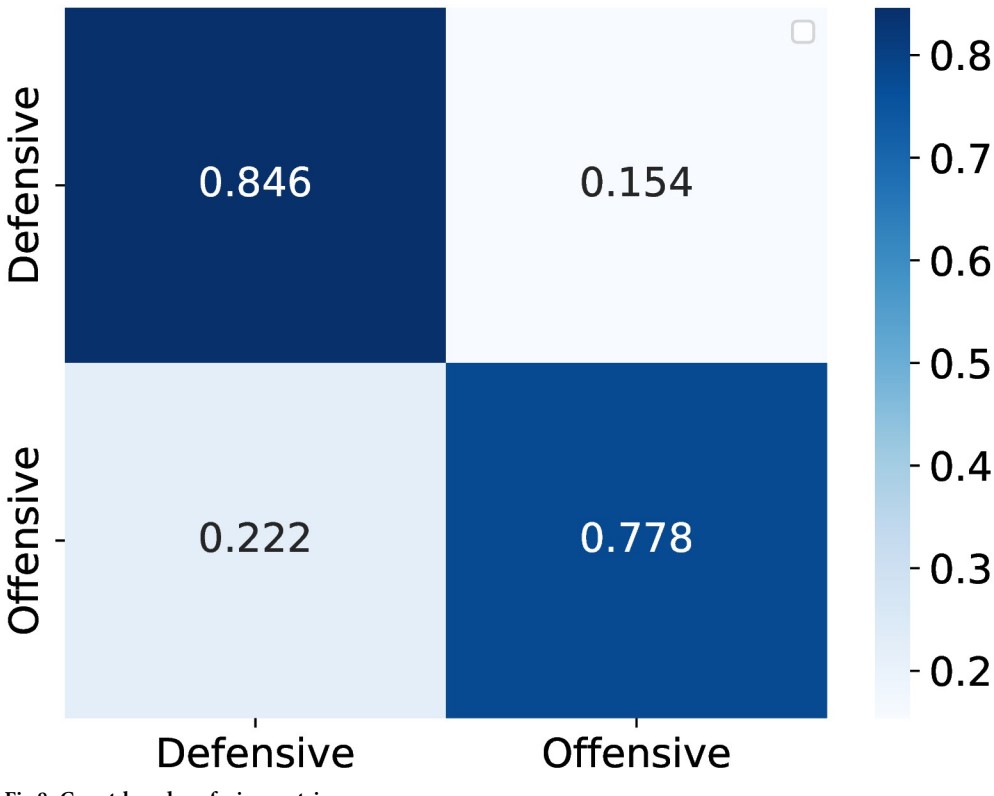

**Fig 8. Count-based confusion matrix.**

These models demonstrated their potential in sports analytics by effectively predicting and interpreting team formation strategies. Future endeavors will explore more sophisticated neural architectures and the integration of additional features, such as player-specific statistics, to enhance the prediction accuracy. The objective is to improve the accuracy of both methodologies by leveraging their respective strengths to provide a more comprehensive understanding of team strategies.

## Conclusion

In this study, we utilized the Elo rating system, rolling regression, autoencoders, various predictive models, and clustering techniques to investigate the strategic formation of soccer teams across five major European leagues. The Elo rating system was used to provide a detailed hierarchy of teams, beyond simple rankings, while rolling regression was used to understand team trends. By combining these two methods, we could discern trends across soccer teams, irrespective of their corresponding league. Additionally, our study explored team formation strategies using two approaches: the count- and ranking-based approaches. Through clustering, we analyzed the similarities among teams in certain situations to determine which teams used similar strategies under specific conditions. We also employed various predictive models to assess the strategy prediction accuracy in specific scenarios. Finally, we analyzed all team strategies using feature extraction with an autoencoder. Because correct labels for offensive or defensive strategies were not available, K-means clustering was used to assign accurate labels to formations. Using a MLP, we predicted team strategies based on clustered labels by considering the team level and trends. The results revealed that count-based predictions were more

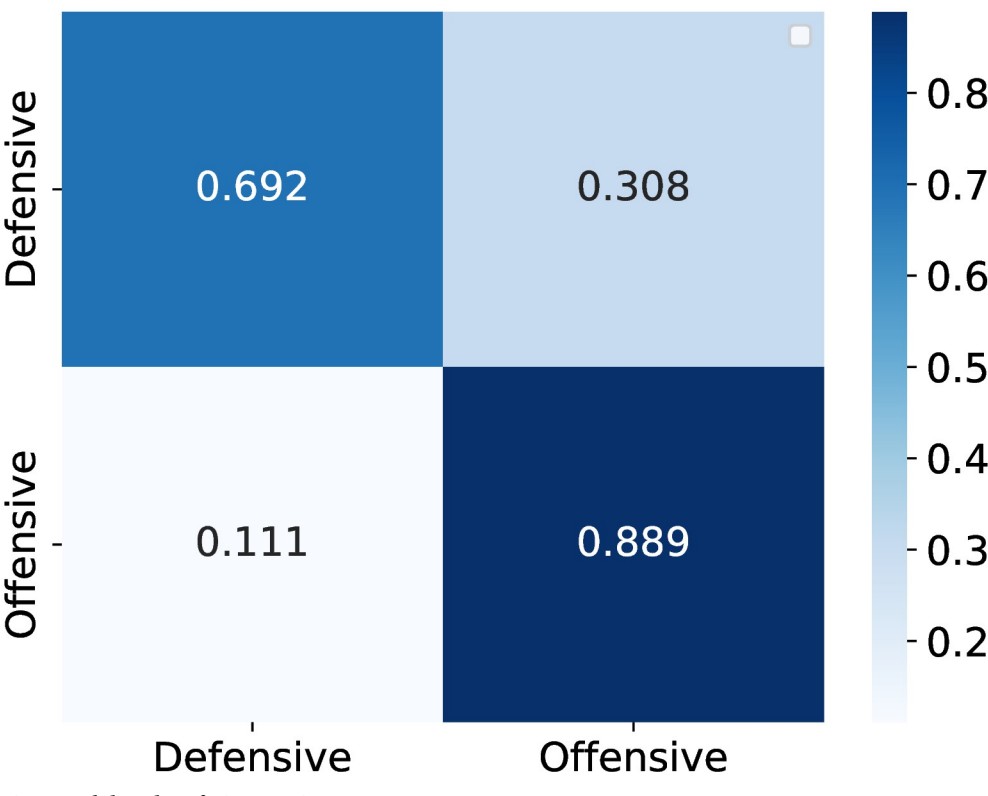

**Fig 9. Rank-based confusion matrix.**

accurate for defensive strategies, while ranking-based predictions were more accurate for offensive strategies. Each prediction method has its advantages and disadvantages, with count-based methods being approximately 15% more accurate in predicting defensive strategies and ranking-based methods being approximately 11% more accurate in predicting offensive strategies. We hope that this study provides a foundation for coaches and strategists to make informed decisions and effectively formulate strategies against certain opponents.

Despite its many advantages, this study has some limitations. The proposed model demonstrated high predictive accuracy but also exhibited several limitations. First, despite measures such as cross-validation and dropout regularization, the relatively small dataset and the complexity of the model raise the possibility of overfitting. Second, the generalizability of this study is constrained as the dataset focuses on a single season of matches from the top five European leagues. This limitation may prevent the full capture of strategic variations across seasons, leagues, or international competitions. Addressing these limitations will be a key focus of future research, which aims to expand the dataset to include multiple seasons and additional leagues, as well as to further evaluate the robustness of the model by employing independent test sets. Future studies should collect data from a wider variety of leagues and teams to address these limitations and undertake more comprehensive analyses. Although we predicted team strategies with relative accuracy, unexpected aspects of team formations or match outcomes were observed. We recognized trends across various games; however, not all trends were utilized in this study. Future work should incorporate all unused trends and seek to design methods for predicting team formations and match outcomes, considering the opposition level. These efforts will aim to predict a broader array of soccer strategies and develop applications for a

larger number of teams, thereby enhancing the global understanding and application of strategic formations in soccer.

## Author Contributions

**Conceptualization:** Dong Hee Jung, Jason J. Jung.

**Data curation:** Dong Hee Jung, Jason J. Jung.

**Formal analysis:** Dong Hee Jung, Jason J. Jung.

**Investigation:** Dong Hee Jung, Jason J. Jung.

**Methodology:** Dong Hee Jung, Jason J. Jung.

**Project administration:** Jason J. Jung.

**Resources:** Dong Hee Jung.

**Software:** Dong Hee Jung.

**Supervision:** Jason J. Jung.

**Validation:** Dong Hee Jung, Jason J. Jung.

**Visualization:** Dong Hee Jung.

**Writing – original draft:** Dong Hee Jung.

**Writing – review & editing:** Jason J. Jung.

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
