## [Decision Letter · Decision Letter 0]

16 Jun 2024

PONE-D-24-17238Data-driven understanding on soccer team tactics and ranking trends: Elo rating-based trends on European soccer leaguesPLOS ONE

Dear Dr. Jung,

Thank you for submitting your manuscript to PLOS ONE. After careful consideration, we feel that it has merit but does not fully meet PLOS ONE’s publication criteria as it currently stands. Therefore, we invite you to submit a revised version of the manuscript that addresses the points raised during the review process.

We look forward to receiving your revised manuscript.

Kind regards,

Julio Alejandro Henriques Castro da Costa

Academic Editor

PLOS ONE

Reviewers' comments:

Reviewer's Responses to Questions

**Comments to the Author**

1. Is the manuscript technically sound, and do the data support the conclusions?

Reviewer #1: Partly

Reviewer #2: Yes

2. Has the statistical analysis been performed appropriately and rigorously? 

Reviewer #1: Yes

Reviewer #2: Yes

3. Have the authors made all data underlying the findings in their manuscript fully available?

Reviewer #1: Yes

Reviewer #2: Yes

4. Is the manuscript presented in an intelligible fashion and written in standard English?

Reviewer #1: No

Reviewer #2: Yes

5. Review Comments to the Author

Reviewer #1: Abstract: Provides good overview, though with some grammatical errors, and lacks details about the methodology.“Using a dataset comprising matches from the top five European soccer leagues, we analyze team performance trends over time using the Elo rating system and rolling regression.” should be “Using a dataset comprising matches from the top five European soccer leagues, we analyzed team performance trends over time using the Elo rating system and rolling regression.”

There are some areas with too few references, and the current references could be more varied and recent.

Introduction: Well-structured, though with the potential for further detail and more recent references.“The main goal of this research is not to directly assist coaches’ decision-making by predicting specific teams’ tactics and game outcomes, but rather to forecast strategies based on team trends, thereby aiding their game preparation process.” should be revised for clarity: “The main goal of this research is to forecast team strategies based on trend analysis, thereby aiding coaches in their game preparation process, rather than directly predicting specific teams’ tactics and game outcomes.”

Add recent references for claims about the impact of big data and AI on sports analytics. Moreover, plos one does not consider reviews.

The manuscript presents an interesting study but requires major revisions to enhance clarity, detail, and academic rigor.

Reviewer #2: Thank you to the authors for producing an insightful manuscript based on interesting and potentially very useful research and statistical analysis techniques.

Overall, the manuscript is well written and detailed, however there are some changes required to prepare this manuscript for publication. Specific points and requests are outlined throughout this document.

Abstract

1. In the following sentence:

‘In this study, the application of these technologies within the domain of association soccer is examined, with a particular focus on predicting team strategies via team trend analysis.’

Would methodologies be a more appropriate term to use instead of technologies?

2. The following sentence does not read correctly and does not make sense:

‘The prediction of strategies from soccer match datasets challenges.’

Please rewrite this sentence.

Introduction

Line 31 – It should read ‘overlook’ not ‘overlooking’ – please change this.

Methodology

There appears to be no title for the methodology section. Please add this.

There is no mention of the statistical program/coding method that was used to calculate all of the models outlined within the study. Please include this.

Line 115 and 119/120 repeat the same information about using the Elo rating system for teams not individuals. This only needs to be included once in this section so one of these sentences should be edited/removed.

Line 141 – the manuscript has already mentioned that Elo was used in chess, so it is not required to be mentioned again here.

Line 188 – The authors have highlighted that the Elo-based approach addresses limitations that other ranking systems have such as the ability to consider opponent strength and the impact of home and away matches. I understand that opponent strength is considered within the RA and RB, but I am unsure how the Elo system considers the impact of home and away matches. I may have missed this point, but please can you clarify this point in the text.

Results

Line 294 – The authors should provide reasons as to why domestic and continental club tournaments do not provide a consistent and reliable set of data, but domestic league games do.

Line 318 – the description of the requirements for each cluster is very vague. The authors should expand on these explanations to provide further insights into the tactics and strategies incorporated into each cluster:

‘For instance, Cluster 2 consisted of teams that adopted widely used strategies, whereas Cluster 1 was characterized by teams that employed less common tactics.’

Line 321 – The description of ‘Cluster 0’ should be moved earlier with the descriptions of cluster 1 and cluster 2.

Table 4 and Table 5 – the different models within these tables have not been described within the methodology section of the paper. A description of any statistical models used within the study should be clearly written for the reader to understand and be able to replicate.

Line 372 – ‘The superior recall of random forest in the ranking-based approach suggests its strength in accurately classifying true-positive instances, indicating its ability to effectively capture nuanced differences in the impact of strategic factors on match outcomes.’

This statement on the effectiveness of the random forest model for rank-based approaches seems to be too strong. This model had a recall of 0.72 which, while it was the best predictor compared to the other models, it still means that it is incorrect 28% of the time. This is not highly accurate, so this sentence needs to be rewritten to reflect that.

Conclusion

Figure 8a and 8b should not appear in the conclusion and need to be moved into the model evaluation section – no more new data should be presented within the conclusion.

The conclusion is well written, clear and concise and summarizes the main findings of the manuscript well.

6. PLOS authors have the option to publish the peer review history of their article (what does this mean?). If published, this will include your full peer review and any attached files.

Reviewer #1: **Yes: **Nadjat Umaru Djagana

Reviewer #2: No

---

## [Author Response · Author response to Decision Letter 0]

2 Jul 2024

We thank the reviewers for their valuable comments. 

All the responds have been carefully mentioned in the attached file.

---

## [Decision Letter · Decision Letter 1]

22 Nov 2024

PONE-D-24-17238R1Data-driven understanding on soccer team tactics and ranking trends: Elo rating-based trends on European soccer leaguesPLOS ONE

Dear Dr. Jung,

Thank you for submitting your manuscript to PLOS ONE. After careful consideration, we feel that it has merit but does not fully meet PLOS ONE’s publication criteria as it currently stands. Therefore, we invite you to submit a revised version of the manuscript that addresses the points raised during the review process.

We look forward to receiving your revised manuscript.

Kind regards,

Julio Alejandro Henriques Castro da Costa

Academic Editor

PLOS ONE

Journal Requirements:

Reviewers' comments:

Reviewer's Responses to Questions

**Comments to the Author**

1. If the authors have adequately addressed your comments raised in a previous round of review and you feel that this manuscript is now acceptable for publication, you may indicate that here to bypass the “Comments to the Author” section, enter your conflict of interest statement in the “Confidential to Editor” section, and submit your "Accept" recommendation.

Reviewer #3: All comments have been addressed

Reviewer #4: All comments have been addressed

2. Is the manuscript technically sound, and do the data support the conclusions?

Reviewer #3: Yes

Reviewer #4: Partly

3. Has the statistical analysis been performed appropriately and rigorously? 

Reviewer #3: Yes

Reviewer #4: Yes

4. Have the authors made all data underlying the findings in their manuscript fully available?

Reviewer #3: Yes

Reviewer #4: Yes

5. Is the manuscript presented in an intelligible fashion and written in standard English?

Reviewer #3: Yes

Reviewer #4: Yes

6. Review Comments to the Author

Reviewer #3: Question 1:Line 13-14，“Despite market expansion, soccer coaches’ training and tactical decisions are often influenced more by personal judgment and experience than by data-driven insights.”This description is too absolute. Is there any literature to support it?

Question 2:The introduction of experimental data sources should be placed in the first half of the paper

Question 3:It is suggested to add the introduction of football game Tactics

Reviewer #4: Dear Author,

The paper addresses an important and emerging field in sports analytics, particularly soccer. By focusing on team trends and strategy prediction, it contributes to advancing the use of data science in sports. The study is well-structured and provides clear research questions. However, there are areas where clarity and depth could be improved, particularly in explaining the methodological approaches and integrating findings with practical applications.

The introduction provides a clear overview of the significance of soccer analytics and highlights the role of big data and machine learning in advancing sports analytics. However, A brief explanation of the Elo rating system and its relevance to team performance could strengthen the context.

The qualitative methodology is suitable for the study, the dual approach of count-based and rank-based methods provides a comprehensive framework for analyzing strategies but more clarity on dataset specifics (e.g., sample size, range of seasons analyzed, preprocessing steps) would enhance replicability.

Results provides quantitative findings, such as the accuracy of predictions (85% for defensive strategies and 89% for aggressive strategies), but Lack of comparative benchmarks with other models or methods makes it difficult to evaluate the study’s innovation.

The discussion highlights the practical implications of predictive models for coaching and game preparation, but it could delve deeper into the limitations of the study, such as potential overfitting or generalizability issues.

References are diverse and relevant, covering foundational concepts and recent advancements in sports analytics, but some references appear dated; more recent studies could be incorporated to reflect current trends in machine learning and sports analytics.

The study’s focus on trend-based strategy prediction is novel and valuable, but the lack of practical implementation diminishes its immediate impact.

The study is a valuable contribution to soccer analytics, but certain areas need refinement, particularly in methodology transparency, results presentation, and discussion depth.

7. PLOS authors have the option to publish the peer review history of their article (what does this mean?). If published, this will include your full peer review and any attached files.

Reviewer #3: No

Reviewer #4: No

---

## [Author Response · Author response to Decision Letter 1]

16 Dec 2024

We thank the reviewers for their helpful comments. All the replies on the review comments have been included in the attached file.

---

## [Decision Letter · Decision Letter 2]

17 Jan 2025

Data-driven understanding on soccer team tactics and ranking trends: Elo rating-based trends on European soccer leagues

PONE-D-24-17238R2

Dear Dr. Jung,

We’re pleased to inform you that your manuscript has been judged scientifically suitable for publication and will be formally accepted for publication once it meets all outstanding technical requirements.

Kind regards,

Julio Alejandro Henriques Castro da Costa

Academic Editor

PLOS ONE

Additional Editor Comments (optional):

Reviewers' comments:

Reviewer's Responses to Questions

**Comments to the Author**

1. If the authors have adequately addressed your comments raised in a previous round of review and you feel that this manuscript is now acceptable for publication, you may indicate that here to bypass the “Comments to the Author” section, enter your conflict of interest statement in the “Confidential to Editor” section, and submit your "Accept" recommendation.

Reviewer #4: All comments have been addressed

2. Is the manuscript technically sound, and do the data support the conclusions?

Reviewer #4: Yes

3. Has the statistical analysis been performed appropriately and rigorously? 

Reviewer #4: (No Response)

4. Have the authors made all data underlying the findings in their manuscript fully available?

Reviewer #4: Yes

5. Is the manuscript presented in an intelligible fashion and written in standard English?

Reviewer #4: Yes

6. Review Comments to the Author

Reviewer #4: General Comment

The manuscript provides a compelling analysis of soccer match outcomes, focusing on the unexpected victories of weaker teams against stronger opponents. The combination of clustering and predictive modeling is well-executed, offering valuable insights into team strategies and match dynamics. The study is methodologically sound, with appropriate use of gradient boosting, logistic regression, and neural networks, alongside detailed clustering analysis. The findings contribute to a growing understanding of how data-driven methods can enhance sports analytics.

The manuscript's strengths lie in its clear articulation of the research goals, comprehensive results analysis, and balanced evaluation of methodologies. Overall, this is a well-constructed and insightful study that holds potential for advancing the field of sports analytics. With minor revisions and additional context, it could make a significant contribution to the literature.

Introduction

• It highlights the increasing adoption of advanced analytics in sports, particularly for optimizing training schedules, injury prevention, and tactical decisions.

• The introduction ties the significance of these technologies to the global popularity and commercialization of soccer.

Methodology

• This methodology offers a robust and nuanced framework for analyzing soccer team performance and strategies. Its adaptability and use of advanced computational techniques ensure its applicability across various competitive leagues.

Results

• This study offers valuable insights into soccer team strategies through robust clustering and prediction methodologies. While the findings demonstrate strong predictive capabilities and nuanced strategic patterns, expanding the dataset and integrating additional features would significantly enhance the depth and applicability of future analyses. The models and methodologies used lay a strong foundation for advancing sports analytics.

Discussion

• The discussion specific model performances, like gradient boosting’s success, and situates this in the context of soccer strategies. This shows an understanding of the interplay between model design and data characteristics.

Bibliography/References

The references cover key areas such as machine learning, sports analytics, clustering techniques, injury prediction, big data applications, and tactical performance analysis, providing a holistic view of the subject matter. However, some entries lack uniform formatting, especially in terms of punctuation, capitalization, and inclusion of complete publication details.

Decision

The manuscript exhibits a strong understanding of the subject and provides a comprehensive analysis of machine learning and sports analytics, particularly in the context of soccer.

Recommendations for Authors:

• Standardize and verify all references for completeness and relevance.

• Align the manuscript’s discussion, results, and references to highlight the study's contributions more clearly.

• Emphasize the practical applications of the research in areas like performance improvement, injury prevention, or strategic planning in soccer.

• Improve the manuscript’s readability and graphical presentation for better engagement with readers.

7. PLOS authors have the option to publish the peer review history of their article (what does this mean?). If published, this will include your full peer review and any attached files.

Reviewer #4: **Yes: **Kalidoss D

---

## [Editor Report · Acceptance letter]

24 Jan 2025

PONE-D-24-17238R2 

PLOS ONE

Dear Dr. Jung, 

I'm pleased to inform you that your manuscript has been deemed suitable for publication in PLOS ONE. Congratulations! Your manuscript is now being handed over to our production team.

Kind regards, 

on behalf of

Dr. Julio Alejandro Henriques Castro da Costa 

Academic Editor

PLOS ONE